# Peripheral Blood T-lymphocyte Phenotypes in Mother-Child Pairs Stratified by the Maternal HPV Status: Persistent HPV16 vs. HPV-Negative: A Case-Control Study

**DOI:** 10.3390/v14122633

**Published:** 2022-11-25

**Authors:** Helmi Suominen, Anna Paaso, Hanna-Mari Koskimaa, Seija Grénman, Kari Syrjänen, Stina Syrjänen, Karolina Louvanto

**Affiliations:** 1Department of Obstetrics and Gynecology, Faculty of Medicine and Health Technology, Tampere University, Arvo Ylpön katu 34, 33520 Tampere, Finland; 2Department of Oral Pathology and Oral Radiology, Institute of Dentistry, Faculty of Medicine, University of Turku, 20014 Turku, Finland; 3Department of Obstetrics and Gynecology, Turku University Hospital and University of Turku, 20014 Turku, Finland; 4SMW Consultants, Ltd., 21620 Kaarina, Finland; 5Department of Pathology, Turku University Hospital, 20014 Turku, Finland; 6Department of Obstetrics and Gynecology, Tampere University Hospital, 33100 Tampere, Finland

**Keywords:** human papillomavirus, HPV, T-lymphocyte, immunophenotyping, mother, child, oral infection, genital infection

## Abstract

Only few studies exist on the phenotype distribution of peripheral blood lymphocytes concerning persistent oral HPV infection. T-lymphocyte subsets were phenotyped in women who had persistent genital or oral HPV16 infection, using HPV-negative women as a reference group. A subset of 42 mothers and their children (*n* = 28), were stratified into two groups according to the mothers’ HPV status. PBMCs from previously cryopreserved venous samples were immunophenotyped by flow cytometry. Proportions of the CD4^+^ or CD8^+^ lymphocytes by their immunophenotype subsets were compared between HPV-positive and -negative mothers and their children. The mean rank distribution of CD8^+^ memory cells was significantly higher among mothers with persistent genital HPV16 infection. The median levels of both the antigen-presenting CD4^+^ cells and activated CD8^+^ cells were significantly lower in mothers with persistent oral HPV16 infection. When oral and genital HPV16-persistors were analyzed as a group, a marker of terminal effector cells was significantly increased as compared to HPV-negative women. Significantly higher levels of activated CD4^+^, CD8^+^ and circulating CD8^+^ memory cells were found among children whose mothers had persistent oral HPV16 infection. Persistent HPV16 infections are associated with changes in peripheral blood T-lymphocyte subsets. The mother’s persistent oral HPV16 infection possibly results in immune alterations in her offspring.

## 1. Introduction

Human papillomavirus (HPV) can be acquired by nonsexual or sexual transmission [1,2,3,4,5,6], resulting in acute, latent, or chronic infection of the epithelial cells. Persistent HPV infections are known to be the main risk factor for the malignant transformation of the epithelial cells resulting in cervical lesions. Typically, most of these HPV-induced lesions regress spontaneously, by mechanisms that are still incompletely elucidated. Persistent HPV infections in the female genital tract can cause genital warts and squamous intraepithelial lesions (SILs) [7]. Certain HPVs are defined as high-risk (HR) types because of their high propensity to cause persistent infections, high-grade SIL (HSIL) and carcinoma [8]. In addition to HR-HPV, interactions of viral and host-related factors are needed in this carcinogenetic process [9].

The host’s immune system is presumed to be a fundamental determinant of the outcome of HPV infection [10]. The adaptive immune system plays an important part in the immune defense against HPV infection along with the innate immune system. The former consists of B-lymphocytes which are mainly responsible for the antibody-mediated immune response and T-lymphocytes which are responsible for the cell-mediated immune response (CMI). When effective, an adaptive immune system contributes to the regression of a persistent HPV infection, whereas a failure to develop sufficient CMI can result in the persistence and progression of an HPV infection [11,12,13,14,15].

Fortunately, most HPV-infected individuals can clear their infection by an efficient CMI response, in which CD4^+^ T cells are of key importance. Previous studies have shown CD4^+^ T-cell-mediated immune response decreases in subjects who develop cancer precursor lesions or cancer due to a persistent HPV infection [11,16,17]. CD4^+^ T cells participate in initiating and maintaining the immune response, whereas CD8^+^ T-cell function as cytotoxic effectors. However, the role of CD8^+^ lymphocytes in HPV infection is less clear than that of CD4^+^ lymphocytes [18]. Regulatory T-lymphocytes (T_regs_) express CD4^+^ and CD25^+^ and because T_regs_ function in suppressing the immune system, these cells are believed to be involved in the progression of persistent HPV infections [19,20]. Compromised CD8^+^ T-cell function has been associated with chronic viral infections, due to the expression of several inhibitory receptors [21,22].

According to current understanding, the CMI system appears to play an important role in the progression or regression of an HPV infection. Immunophenotyping of peripheral blood T cells could cast further light on understanding the outcomes (persistence, clearance) of HPV infections even without any clinical HPV lesions. The prospective Finnish Family HPV Study (FFHPV) cohort was used to make comprehensive immunophenotyping of peripheral blood T-lymphocyte subsets in mothers with persistent HPV16 infection, accompanied by similar analyses of their children.

## 2. Materials and Methods

### 2.1. Study Cohort

The Finnish Family HPV (FFHPV) study is a longitudinal cohort study that was designed to clarify the dynamics of HPV infections within families. Since its onset in 1998, the study has been conducted jointly by the Department of Obstetrics and Gynecology, Turku University Hospital and the Institute of Dentistry, Faculty of Medicine, University of Turku. The original study protocol and its amendments were approved by the Research Ethics Committee of Turku University Hospital (#3/1998, #2/2006 and 45/180/2010).

Altogether, 329 pregnant women in their third trimester and their newborns were enrolled in the study between 1998 and 2001, based on written informed consent of all participants. The study design and the key characteristics of the participants have been detailed in a series of previous reports [19,23,24,25]. In the present analysis, a subset of mother–child pairs (42 mothers and 28 children) was used, selected from the families specially followed-up for HPV16-specific (CMI) responses, as previously described [23,26].

### 2.2. Samples

Genital and oral scraping samples for HPV genotyping were taken with a cytobrush at the baseline and during the follow-up visits: at day 3 (before leaving the hospital) and at 1-, 2-, 6-, 12-, 24-, 36-month- and 6-year follow-up visits. The samples were fixed and immediately frozen and stored at −70 °C. HPV DNA was extracted by using the high salt method, as previously described [1]. HPV genotyping was performed by using the Multimetrix Kit (Progen Biotechnik GmbH), which detected 24 low-risk (LR-HPV) and high-risk (HR-HPV) HPV genotypes (HPV6, 11, 42, 43, 44, 70, 26, 53, 66, 16, 18, 31, 33, 35, 39, 45, 51, 52, 56, 58, 59, 68, 73, 82 [27]).

### 2.3. Flow Cytometric Analysis

Peripheral blood mononuclear cells (PBMCs) from previously cryopreserved venous samples of the mothers and their children were used. The participants were recalled for blood sampling, the mean age of the children being 12.2 years [19]. A detailed description of blood sample collection and processing has been published previously [19,23,24,25,26,28]. PBMCs were stained in 96-well U-bottom plates in the concentration of 200,000–500,000 cells/well. The plate was centrifuged for 5 min with 450 g at 4 °C followed by a washing step with FACS buffer (PBS/0.5% BSA/2 mM EDTA) and centrifuged again. After the supernatants were removed, the diluted antibody mix (Table 1) was added to samples (50 μL/well) and incubated for 30 min (dark, 4 °C). After washing again with FACS buffer, the supernatant was removed and resuspended to 5 μL of CellFix (BD Bioscience). Samples were analyzed within 2 h in a BD 4-lasers LSR Fortessa™ cell analyzer (BD Bioscience, NJ, USA) and gating was performed with Flow-Jo Software (BD Bioscience, New York, NJ, USA).

T-cell subpopulations were defined by the presence or absence of different cell surface markers to (i) either early or late activated cells, (ii) memory or naïve cells, as well as (iii) differentiated or undifferentiated cells, closely following the protocol reported by Rodriguez et al. [16]. The T-cell subpopulations in this study were defined as follows: (1) Early-activated T cells: CD4^+^CD69^+^ and CD8^+^CD69^+^; (2) Late-activated T cells: CD4^+^CD25^+^, CD8^+^CD25^+^, CD4^+^HLA-DR^+^, CD8^+^HLA-DR^+^, CD4^+^CD38^+^ and CD8^+^CD38^+^; (3) Memory T-cells: CD45RO^+^CD45RA^−^; (4) Naïve T cells: CD45RO^−^CD45RA^+^; (5) Central memory (CD45RO^+^CCR7^+^) and -naïve cells (CD45RA^+^CCR7^+^); (6) Effector-memory (CD45RO^+^CCR7^-^) and-naïve cells (CD45RA^+^CCR7^-^); (7) Resting memory cells and differentiated naïve cells: CD27^+^CD45RO^+^ and CD27^+^CD45RA^+^; (8) Differentiated memory cells and undifferentiated naïve cells: CD57^+^CD45RO^+^ and CD57^+^CD45RA^+^.

CD69 markers in CD4^+^ and CD8^+^ T cells are reported as percentages of the total number of lymphocytes, the lymphocyte population being selected based on the forward scatter/side scatter pattern. The CD45RO, CD45RA, CD27 and CD57 markers are reported as percentages of CD3^+^CD4^+^ or CD3^+^CD8^+^ lymphocytes. HLA-DR and CD38 marker populations were estimated as a percentage of the CD3^+^CD4^+^ or CD3^+^CD8^+^lymphocytes. The CD57 subpopulation was estimated as a percentage of CD3^+^CD4^+^ or CD3^+^CD8^+^ lymphocytes that were CD27^+^CD45RA^+^ or CD27^+^CD45RO^+^.

### 2.4. Statistical Analyses

To be eligible for the present analysis, the mother had persistent HPV16 infection either in the genital or oral mucosa. As a reference group, the mothers who tested constantly HPV-negative during the follow-up time were selected as described earlier [23,26]. In addition to the mothers in these two groups, the children were subjected to similar analyses. Altogether, a group of 42 mothers and their children (*n* = 28) was divided into four subgroups based on the mothers’ HPV infection status: Group (1) the first group included 10 mothers, who developed an incident CIN with persistent genital HPV16 infection during the follow-up and 10 children of these mothers; Group (2) the second group consisted of 7 mothers with persistent oral HPV16 infection and their 7 children. Persistent HPV16 infection (genital or oral) was defined by testing HPV16 positive at least in two (or more) consecutive follow-up visits. As to the reference groups, Group (3), consisted of 20 mothers who tested genital-HPV-negative during the follow-up and 8 children of these mothers; Group (4) consisted of 5 mothers who tested repeatedly oral-HPV-negative and their 3 children. To be eligible in these reference groups, the mother had to be always HPV-negative, with no HPV-positive genital or oral sample during the follow-up time.

In statistical analysis, the two HPV16-positive groups (oral and genital) were combined as the group of HPV16 carriers, to be compared with the combined group of always HPV-negatives. Between all the groups above, differences in the mothers’ mean age, follow-up time, gender of the children as well as oral HPV status of the children were assessed. The proportion of the CD4^+^ or CD8^+^ lymphocytes by their immunophenotype subsets were compared between the groups described above; first between the mothers, secondly between the children, and across the mother–child pairs. Bonferroni post hoc tests were used to control for multiple comparisons. Bonferroni uses *t*-tests to perform pairwise comparisons between group means but controls the overall error rate by setting the error rate for each test to the experiment-wise error rate divided by the total number of tests. Hence, the observed significance level is adjusted for the fact that multiple comparisons are being made. In null hypothesis testing by this post hoc test we used the same significance level (alpha) as the settings in options. All statistical analyses were performed by using SPSS statistical software (25.0). All tests were performed two-sided, and statistical significance was declared at the *p*-value < 0.05.

## 3. Results

The baseline characteristics of the subgroups of the 42 mothers and their children (*n* = 28) included in the present analysis are shown in Table 2. The mothers’ mean age ranged between 37 and 40 years and their children’s mean age ranged between 12.2 and 14.7 years. From the 28 children included in this study, a total of 17 children had an oral HPV infection recorded during the follow-up, of which three had a persistent oral HPV infection.

Table 3 and Table 4 summarize the results of the CD4^+^ and CD8^+^ immunophenotypic subgroups stratified by the mother–child dyads, using 23 and 25 CD4^+^ and CD8^+^ T-cell surface markers, respectively. The highest proportion of T-lymphocytes (CD3^+^ T cells; presented as median percentages of cells among PBMCs) was seen in the mothers having persistent genital HPV16 infection (73.0%), while the lowest median proportion (47.0%) was detected in the mothers with persistent oral HPV infection (Table 3). T-lymphocyte counts were significantly different only between the mothers with persistent genital HPV16 infection and their HPV-negative controls (*p* = 0.019). In children, the distribution of CD3^+^ T cells was similar across the three comparison groups. No statistically significant differences were found in the distribution of the CD4^+^ (helper cells) and CD8^+^ (cytotoxic) T cells within the CD3^+^ T cells between HPV^+^ or HPV^−^ mothers or the respective groups of children (Table 3 and Table 4).

Among mothers, only one significant difference between HPV16 carriers and HPV-negative women was seen in the frequency of immunophenotypic subsets of CD3^+^CD4^+^ T cells; the level of HLADR^+^CD3^+^CD4^+^ (a marker for T-cell activation) was lower in the mothers with persistent oral HPV16 (*p* = 0.038). The same was true with the children: children of the mothers with persistent oral HPV16 infection had higher levels of CD38^+^HLADR^+^CD3^+^CD4^+^ cells; a marker of T-cell activation (*p* = 0.038).

When the immunophenotypic subsets of CD3^+^CD8^+^ cells were compared (Table 4), the levels of HLADR^+^CD3^+^CD8^+^, CD38^+^HLADR^+^CD3^+^CD8^+^ and CD38^−^HLADR^+^CD3^+^CD8^+^ subsets were significantly higher in children of the mothers having persistent oral HPV16 infection as compared with the HPV-negative counterparts; *p* = 0.006, *p* = 0.008 and *p* = 0.018, respectively. In addition, the CD45RO^+^CD8^+^ subset (a marker of memory T cells), was significantly higher among the children of the mothers with persistent oral HPV16 infection as compared with the HPV-negative counterparts.

Supplemented statistical analysis was also further performed for the control of multiple comparisons by using the classical Bonferroni test (BT). When BT was used, the three *p*-values that had been declared significant in Table 3, underwent the following changes: (1) CD3^+^ lymphocytes: Mothers persistent HPV16 infection vs. always HPV-negative, original *p* = 0.019, changed after BT to *p* = 0.084; (2) HLADR^+^CD3^+^CD8^+^: Mothers oral persistent HPV16 infection vs. always HPV-negative, original *p* = 0.038, changed after BT to *p* = 0.668; (3) CD38^+^HLADR^+^CD3^+^CD4^+^: Children of mothers with oral persistent HPV16 vs. always HPV-negative, original *p* = 0.038, changed after BT to *p* = 1.000. When BT was used in Table 4 five *p*-values that were declared significant underwent the following changes: (1) CD3^+^ lymphocytes: Mothers persistent genital HPV16 infection vs. always HPV-negative, original *p* = 0.019, changed after BT to *p* = 0.084; (2) HLADR^+^CD3^+^CD8^+^: Children of mothers with persistent oral HPV16 infection vs. always HPV-negative, original *p* = 0.006, changed after BT to *p* = 1.000; (3) CD38^+^HLADR^+^CD3^+^CD8^+^: Children of mothers with persistent oral HPV16 infection vs. always HPV-negative, original *p* = 0.008, changed after BT to *p* = 1.000; (4) CD38^−^HLADR^+^CD3^+^CD8^+^: Children of mothers with persistent oral HPV16 infection vs. always HPV-negative, original *p* = 0.018, changed after BT to *p* = 1.000; and (5) CD45RO^+^CD8^+^: Children of mothers with persistent oral HPV16 infection vs. always HPV-negative, original *p* = 0.033, changed after BT to *p* = 1.000.

Table 5 depicts the immunophenotypic CD4^+^ and CD8^+^ T-cell subsets stratified by the defined groups. This was conducted because the number of mothers defined by their HPV status was limited. Of the immunophenotypic subsets of CD3^+^CD4^+^ T cells, only the HLADR^+^CD3^+^CD4^+^ T-cell subset was significantly lower (*p* = 0.038) in mothers with persistent oral HPV16 infection than in HPV-negative mothers.

Of the immunophenotypic subsets of CD3^+^CD8^+^ T cells, only one statistically significant difference was found: CD45RO^+^CCR7^−^CD8^+^ cell population (a marker of memory cells), was significantly more abundant among mothers with persistent genital HPV16 infection than in HPV-negative women (*p* = 0.048). This difference to HPV-negative mothers remained significant also when the oral- and genital HPV16-positive mothers were pooled together, (*p* = 0.033).

Among the mothers with persistent oral HPV16 infection, CD38^+^HLADR^+^CD3^+^CD8^+^ cells (a marker of activated T cells) were significantly lower (mean 3.67 ± SD 2.18) than in HPV-negative mothers (mean 7.70 ± SD3.61; *p* = 0.036). When the oral and genital HPV16 persisters were pooled, the CD45RA^+^CCR7^−^CD8^+^ cell population (a marker of terminal effector cells), was significantly increased in HPV16-positive mothers as compared with their HPV-negative mothers (*p* = 0.033).

Finally, in children, significant differences in the levels of CD4^+^ and CD8^+^ T-cell subsets were recorded only between those in the oral (but not genital) HPV16-persistor group and their control group (i.e., children born to mothers who always tested HPV-negative in their oral samples) as seen in Table 6. HLADR^+^CD3^+^CD4^+^ CD8^+^ cells (a marker of both CD4^+^ and CD8^+^ activated T cells; MHC II^+^ T cells) were significantly higher in children whose mothers had persistent oral HPV16 infection (*p* = 0.018). Similarly, a significant increase in the number of activated T cells was found both for CD4^+^ helper cells (CD38^+^HLADR^+^CD3^+^CD4^+^) (*p* = 0.038) and CD8^+^ suppressor cells (CD38^+^HLADR^+^CD3^+^CD8^+^) (*p* = 0.008). Additionally, the levels of HLADR^+^CD3^+^CD8^+^ cells (i.e., activated T-suppressor cells), were significantly higher (*p* = 0.005) as were the levels of CD45RO^+^ CD8^+^ cells (*p* = 0.033), i.e., a population of circulating memory cells.

## 4. Discussion

In the present study, we analyzed the distribution of T cells and their subpopulations in the peripheral blood of mothers with either persistent oral or genital HPV16 infection using constantly HPV-negative mothers as the reference group. Similar analyses were also performed in the children of these mothers because immunological recognition of HPV16 seems to occur in early childhood [19,26,28,29].

These analyses demonstrate that the mothers with persistent genital HPV16 infection had higher levels of CD3^+^ lymphocytes and effector memory CD8^+^ T cells (CD45RO^+^CCR7^−^) in their blood as compared with always HPV-negative mothers. When all HPV16-positive mothers were pooled together, they had also higher levels of terminally differentiated (T_EMRA_) CD8^+^ cells (CD45RA^+^CCR7^−^), which are the most effective CD8^+^ cells in destroying tumor cells and virus-infected cells [30,31,32]. Mothers with persistent oral HPV16 had significantly lower levels of activated helper (CD4^+^) and suppressor (CD8^+^) T cells among the CD3^+^ lymphocyte population.

As to other viruses, a longitudinal study on EBV-infected patients revealed differences in lytic versus latent epitope-specific composition of the CD8^+^ T-cell population in the chronic carrier stage of the infection [33,34]. Latent epitopes acquired CD45RA in the persistent phase of EBV infection, as also found here for HPV16-persistors [33,34]. In line with our results, there are earlier studies reporting differences between naïve (CD45RA^+^) and memory T-cell (CD45RO^+^) populations in peripheral blood in women with HPV infection [11,35,36]. Contradictory to our results, however, Rodriguez et al. reported that CD45RO^+^CD27^−^CD8^+^ T cells in PBMCs were positively associated with HPV persistence whereas CD45RO^+^CD27^+^CD4^+^ T cells had a negative association [16]. However, there is recent evidence suggesting that local CD45RA^+^/CD45RO^+^ expression in cervical intraepithelial lesions (CINs) might even be a prognostic biomarker because the expression is increasing in parallel with the increasing grade of CIN [37].

We also found that mothers with persistent oral HPV16 infection had lower levels of peripheral HLADR^+^CD3^+^CD4^+^ and CD38^+^HLADR^+^CD3^+^CD8^+^ T cells as compared with their HPV-negative counterparts. HLA-DR is a histocompatibility antigen from the MHC II family and both CD4^+^ helper and CD8^+^ cytotoxic T cells bear HLA-DR molecules as important surface activation markers. Both HLA-DR and CD38 molecules are present on immature T- and B- lymphocytes and are re-expressed during an immune response. Antigen-presenting cells are essential for HLA-DR on CD3^+^CD4^+^ T cells. Co-expression of HLA-DR and CD38 is a key marker of CD8^+^ T-cell immune activation during several viral infections, e.g., influenza, Dengue, Ebola, and HIV-1 [38].

Contradictory to our results, Rodrigues and coworkers reported a positive correlation between HLADR^+^CD3^+^CD4^+^ T cells and persistent genital HPV infection [16]. Additionally, Papasavvas et al. found that irrespective of the presence of CIN, HR-HPV-positive women had higher levels of circulating CD38^+^ T cells, yet these women had also HIV infection [39]. Importantly, the presence of the human MHCII isotype, HLA-DR, potentially also identifies a regulatory T-cell population. Regardless of an endogenous expression or a protein acquisition, MHCII on T cells has mainly been associated with the induction of down-regulatory signals in the responding T cell [40,41,42,43,44]. It has also been associated with active rather than resting regulatory T cells (T_regs_) [45]. However, several studies also demonstrate that the MHCII^+^ T cells can activate other T cells [38,46]. In line with the multipotential function of HLA-DR expressing T cells, it has been shown that activated CD8^+^ T cells co-expressing of HLA-DR and CD38 accumulated over a prolonged period in HIV-infected lymphopenia patients. Several studies have also indicated that CD38^+^HLA^−^DR^+^CD8^+^ T cells could play a recovery role in activating immunity and eliminating the virus [37,47].

Our contradictory results above compared to the study by Rodrigues and coworkers might also be due to their studies’ lack of correlation for multiple comparisons [16]. Rodrigues and coworkers also compared two groups as factor variables and T cell populations as dependent variables but no correlation was observed for multiple comparisons which might be a potential issue of obtaining false-positive results [16]. Our setting is fortunate in that we had more than two groups of factor variables (both mothers and children), which enables us to use the conventional methods for compensating for the multiple testing. When the Bonferroni test (BT) was used for the correlation of multiple comparisons, none of the significant comparisons stayed significant, as shown in Table 3 and Table 4. However, the proper use of different post hoc tests for the control of multiple comparisons has been extensively discussed in the statistical literature [48]. Several different MCTs (multiple comparison tests) are available, and there is no unanimous agreement on which is the most suitable one for each setting. In this study, we used BT, which is the most often used of all post hoc tests. However, BT also has disadvantages since it is unnecessarily conservative with weak statistical power. The adjusted α (alpha) is often smaller than required, particularly if there are many tests and/or the test statistics are positively correlated. Therefore, this method often fails to detect real differences. More liberal methods exist, for example, Fisher’s least significant difference (LSD), which does not control the family-wise error rate. In fact, we also tested LSD as the MCT in our calculations, and indeed, the inflation of alpha level was markedly less, around *p* = 0.500 or less in most instances where BT resulted in *p* = 1.000. Considering all the above, our results will need to be further replicated with larger study settings so that we can firmly confirm the findings of this study and the previous studies on this topic.

Interestingly, the children of the mothers with persistent oral HPV16 infection had entirely opposite changes in the proportions of CD38^+^HLADR^+^CD3^+^CD4^+^, HLADR^+^CD3^+^CD8^+^, CD38^+^HLADR^+^CD3^+^CD8^+^ and CD38^−^HLADR^+^CD3^+^CD4^+^CD8^+^ T cells in peripheral blood, being significantly elevated as contrasted to their mothers with declined levels of these subsets. To the best of our knowledge, this is the first study immunophenotyping the peripheral blood lymphocyte subpopulations among mothers with persistent HPV16 infection and their children. Significant alterations in the distribution of lymphocyte subsets were found only in the children whose mothers had persistent oral HPV16 infection. Activated CD3^+^, CD4^+^ and CD8^+^ T cells expressing HLA-DR with or without CD38 were significantly more abundant in these children, compared with children of HPV-negative mothers, being a sign of activated immunity. Importantly, HPV-specific T-cell response was not analyzed in the present study. However, based on our previous data from the Finnish Family HPV Study, we know that the mother seems to be the main HP transmitter to her offspring via mouth [19,20]. In line with this, the present observations could indicate a continuous exposure of the offspring to maternal oral HPV, resulting in T-cell activation. This is supported by the fact that also the effector memory CD8^+^ cells (CD45RO^+^) were expanded in these children. It has been suggested that persistent HPV infection could be associated with undifferentiated memory CD4^+^ T cells and other T-cell subsets that are incompetent in eliminating the viral pathogen [11,16,49,50]. In the present series, CD4^+^ T cells expressing HLADR and CD3 were increased both in mothers with persistent oral HPV16 infection and their children (Table 5 and Table 6).

Despite its unique design, the present study has also limitations. First, the number of study subjects is limited, including only 42 mothers and 28 of their children. When stratified into cases and controls, this small number is the key limiting factor to reaching statistical power. In addition, the correlation for multiple comparisons needs to be considered as this might potentially help in obtaining false-positive results as discussed above. Similarly, the temporal relationship between HPV acquisition and collection of the PBMCs is not known. Therefore, the dynamics of adaptive immunity cannot be thoroughly investigated from these data. Finally, as previously affirmed [16], it is not known whether a persistent HPV infection causes the activation of different T-cell subsets or whether it is the activation of these T-cell subsets that causes the persistence of HPV infection.

In general, a detailed dissection of the T-cell immunophenotypes and their relation to the known outcomes of HPV infections could ideally offer important predictive insights in clinical practice. Thus far, the number of studies assessing HPV immunity in relation to natural history is limited. As there is no reliable way to trace the time of the first HPV exposure, the studies on HPV immunology suffer from this inherent handicap. This study is, to our knowledge, the first to conduct T-cell immunophenotyping in mothers and their children in a longitudinal setting. Taken together, these data suggest that both genital and oral HPV16 infections in mothers are associated with alterations in the relative distribution of peripheral blood T-cell subsets. In children of these women, such alterations in T-cell subsets were only found when the mother had persistent oral HPV16 infection.

## Figures and Tables

**Table 1 viruses-14-02633-t001:** Antibodies used for immunological phenotyping of peripheral blood mononuclear cells by flow cytometry.

Cell Marker	Dilution	Fluorochrome	Antibody Clone	Source
CD3	1/15	APC-Cy7	SK7	Biolegend
CD4	1/15	PerCP-Cy5.5	OKT4	Biolegend
CD8	1/15	FITC	SK1	Biolegend
CD25	1/15	Alexa 700	M-A251	BD Bioscience
CD27	1/15	APC	L128	BD Bioscience
CD45RA	1/15	BV50/BV510	HI100	BD Bioscience
CD45RO	1/15	PE	UCHL1	Biolegend
CD57	1/33	PE Dazzle	HNK-1	Biolegend
CD38	1/15	BV605	HB7	BD Bioscience
CD69	1/15	BV421	FN50	BD Bioscience

**Table 2 viruses-14-02633-t002:** Baseline characteristics of the subgroups of mothers (*n* = 42) and their children (*n* = 28) from the Finnish Family HPV cohort.

		Genital HPV16 Infection		Oral HPV16 Infection	
		Incident ≥ CIN+ with Persistent * Infection	Always Negative	Persistent * Infection	Always Negative
**Mothers**	*N*	10	20	7	5
**Children**	*N*	10	8	7	3
Mean age	Mothers	37.0	40.0	38.7	38.7
	Children	12.2	12.3	14.7	14.7
Gender of the children	Girls	3	4	4	2
	Boys	7	4	3	1
Oral HPV status of the children	Always negative	5	3	3	0
	Incident	1	3	4	2
	Persistent	2	0	0	1
	Fluctuation	1	2	0	0
	Clearance	1	0	0	0
	Incident				
	Persistent				

* Persistent is defined as two or more consequent follow-up visits HPV16 positive. Abbreviations: CIN: cervical intraepithelial neoplasia.

**Table 3 viruses-14-02633-t003:** CD4+ T-cell immunophenotypic subset distribution of the mother–child pairs stratified according to mother’s genital and oral HPV status. Significant median comparisons between the subgroups are bolded.

		HPV16 Infection Status of the Mother					
		Genital HPV16 Infection		Oral HPV16 Infection		Combined HPV16 Infection	
		Persistent * Infection	Always HPV Negative	Persistent * Infection	Always HPV Negative	Persistent * Infection	Always HPV Negative
Marker		Median (%)		Median (%)		Median (%)	
CD3^+^ lymphocytes	Mothers	**73.00 ^a^**	**54.65 ^a^**	47.10	60.10	58.30	54.70
	Children	72.65	67.40	54.50	67.80	67.10	67.80
CD3^+^CD4^+^	Mothers	47.50	51.45	36.60	46.90	44.20	51.10
	Children	40.15	36.90	38.20	41.30	39.50	37.10
CD69^+^CD4^+^	Mothers	1.89	1.54	1.54	0.58	1.72	1.39
	Children	0.39	0.41	0.40	0.50	0.40	0.49
CD25^+^CD4^+^	Mothers	0.13	0.15	0.10	0.10	0.11	0.14
	Children	0.25	0.25	0.27	0.25	0.25	0.25
CD27^+^CD4^+^	Mothers	88.90	90.25	82.30	86.00	88.15	89.90
	Children	92.30	89.00	92.30	95.10	92.30	93.40
HLADR^+^CD3^+^CD4^+^	Mothers	4.92	4.69	**4.27 ^b^**	**6.28 ^b^**	4.45	4.73
	Children	3.06	3.53	3.92	1.81	3.60	2.92
CD38^+^CD3^+^CD4^+^	Mothers	53.00	49.65	49.30	34.50	51.25	48.70
	Children	65.70	67.10	63.00	68.80	65.30	67.20
CD38^+^HLADR^+^CD3^+^CD4^+^	Mothers	2.59	2.74	2.71	3.63	2.70	2.90
	Children	2.12	2.71	**2.70 ^c^**	**1.06 ^c^**	2.37	1.80
CD38^−^HLADR^+^CD3^+^CD4^+^	Mothers	2.33	2.42	1.91	4.29	2.07	2.45
	Children	1.54	1.26	1.46	0.72	1.47	1.11
CD45RA^+^CD4^+^	Mothers	65.20	58.95	59.90	46.00	63.30	58.00
	Children	71.60	72.25	68.10	70.50	69.20	70.50
CD45RA^+^CD27^+^CD4^+^	Mothers	93.60	92.50	88.30	95.60	92.30	92.70
	Children	96.50	94.55	97.20	99.10	96.90	97.60
CD45RA^+^CD27^−^CD4^+^	Mothers	6.37	7.35	11.30	4.37	7.66	7.05
	Children	3.47	5.42	2.73	0.88	3.04	2.30
CD45RA^+^CD57^+^CD4^+^	Mothers	9.18	5.35	5.28	2.37	8.01	5.00
	Children	1.97	4.76	0.96	0.73	1.79	1.69
CD45RA^+^CD57^−^CD4^+^	Mothers	90.80	94.60	94.70	97.60	92.00	94.90
	Children	98.00	95.25	99.00	99.30	98.20	98.30
CD45RA^+^CD57^+^CD27^+^CD4^+3^	Mothers	2.51	0.79	1.35	0.72	1.68	0.78
	Children	0.80	0.86	1.02	0.82	0.91	0.82
CD45RA^+^CD57^−^CD27^+^CD4^+3^	Mothers	86.50	91.25	87.90	94.10	87.00	91.30
	Children	95.85	93.35	96.00	98.30	96.00	96.70
CD45RO^+^CD4^+^ memory	Mothers	41.80	52.15	51.10	63.80	45.25	54.40
	Children	40.10	37.35	36.50	36.80	38.50	37.30
CD45RO^+^CD27^+^CD4^+^	Mothers	80.40	83.60	76.30	79.50	77.80	82.50
	Children	79.25	74.75	80.10	89.40	79.50	83.00
CD45RO^+^CD27^−^	Mothers	19.60	16.40	23.70	20.50	22.20	17.50
	Children	20.75	25.25	19.90	10.60	20.50	17.00
CD45RO^+^CD57^+^CD4^+^	Mothers	13.70	6.71	11.20	4.59	12.45	6.56
	Children	6.46	12.42	5.70	4.06	6.16	5.32
CD45RO^+^CD57^−^CD4^+^	Mothers	86.30	93.30	88.80	95.40	87.55	93.40
	Children	93.55	87.60	94.30	95.90	93.80	94.70
CD45RO^+^CD57^+^CD27^+^CD4^+^	Mothers	3.01	2.30	2.23	2.39	2.72	2.37
	Children	3.75	2.92	4.29	3.21	3.96	3.21
CD45RO^+^CD57^−^CD27^+^CD4^+^	Mothers	77.20	79.55	75.00	78.10	76.00	78.10
	Children	77.10	69.50	77.20	87.90	77.20	79.80

* Mothers developed Incident ≥ CIN+ during the follow-up. *p*-values = ^a^ 0.019, ^b^ 0.038, ^c^ 0.038.

**Table 4 viruses-14-02633-t004:** CD8^+^ T-cell immunophenotypic subset distribution of the mother–child pairs stratified according to mother’s genital and oral HPV status. Significant median comparisons between the subgroups are bolded.

		HPV16 Infection Status of the Mother					
		Genital HPV16 Infection		Oral HPV16 Infection		Combined HPV16 Infection	
		Persistent Infection *	Always Negative	Persistent infection *	Always Negative	Persistent Infection *	Always Negative
**Marker**		**Median (%)**		**Median (%)**		**Median (%)**	
CD3^+^ lymphocytes	Mothers	**73.00 ^a^**	**54.65 ^a^**	47.10	60.10	58.30	54.70
	Children	72.65	67.40	54.50	67.80	67.10	67.80
CD3^+^CD8^+^	Mothers	22.10	18.35	24.40	25.70	22.10	19.80
	Children	25.15	25.30	20.50	20.50	22.20	20.50
CD69^+^CD8^+^	Mothers	1.62	1.83	1.02	3.22	1.46	1.87
	Children	0.76	0.77	0.92	0.93	0.83	0.93
CD25^+^CD8^+^	Mothers	0.06	0.15	0.085	0.33	0.08	0.19
	Children	0.08	0.22	0.12	0.13	0.09	0.17
CD27^+^CD8^+^	Mothers	73.30	70.15	49.50	68.60	64.85	70.00
	Children	84.65	74.05	86.90	93.90	86.50	85.30
HLADR^+^CD3^+^CD8^+^	Mothers	5.65	8.35	6.41	7.74	6.03	8.10
	Children	6.01	7.44	**10.30 ^b^**	**3.80 ^b^**	8.41	5.36
CD38^+^CD3^+^CD8^+^	Mothers	37.80	40.65	36.30	37.50	36.80	37.50
	Children	42.65	52.45	50.50	57.90	49.90	53.50
CD38^+^HLADR^+^CD3^+^CD8^+^	Mothers	2.99	4.42	3.94	5.59	3.27	5.03
	Children	2.92	4.09	**5.63 ^c^**	**2.91 ^c^**	5.07	3.46
CD38^-^HLADR^+^CD3^+^CD8^+^	Mothers	3.40	3.53	3.37	2.80	3.39	3.35
	Children	4.05	3.56	**4.38 ^d^**	**1.20 ^d^**	4.38	2.32
CD45RA^+^CD8^+^	Mothers	79.20	77.55	80.40	73.20	79.80	77.50
	Children	85.35	90.95	87.80	91.30	86.80	91.30
CD45RA^+^CCR7^−^CD8^+^	Mothers	93.80	92.70	95.90	93.90	93.80	93.40
	Children	94.75	96.70	92.90	92.90	94.20	96.60
CD45RA^+^CD27^+^CD8^+^	Mothers	67.70	68.55	42.30	64.20	58.60	68.20
	Children	82.90	73.50	86.90	94.00	86.30	84.00
CD45RA^+^CD27^−^CD8^+^	Mothers	31.30	30.50	57.30	34.90	41.10	30.50
	Children	17.10	26.40	13.00	5.89	13.50	16.00
CD45RA^+^CD57^+^CD8^+^	Mothers	43.60	36.30	53.20	37.20	46.35	36.90
	Children	20.90	30.45	25.90	11.40	25.10	23.10
CD45RA^+^CD57^+^CD27^+^CD8^+^	Mothers	14.20	10.01	6.30	7.60	11.75	9.72
	Children	9.38	8.53	8.03	7.18	8.03	7.18
CD45RA^+^CD57^−^CD8^+^	Mothers	56.30	63.50	46.70	62.70	53.55	62.70
	Children	72.20	69.50	74.10	88.60	74.10	76.90
CD45RA^+^CD57^−^CD27^+^CD8^+^	Mothers	46.50	56.55	36.60	55.30	44.95	56.40
	Children	69.15	65.35	71.90	86.50	71.70	71.70
CD45RO^+^CD8^+^	Mothers	50.20	54.50	46.20	49.60	48.15	49.60
	Children	43.30	38.45	**33.20 ^e^**	**23.00 ^e^**	35.10	32.00
CD45RO^+^CCR7^+^CD8^+^	Mothers	8.69	12.65	9.90	12.00	9.30	12.00
	Children	6.05	7.17	8.72	6.28	7.29	6.54
CD45RO^+^CCR7^−^CD8^+^	Mothers	91.30	87.35	90.10	88.00	90.70	88.00
	Children	93.95	92.85	91.30	93.70	92.70	93.50
CD45RO^+^CD27^+^CD8^+^	Mothers	63.80	61.75	48.80	65.90	60.30	63.00
	Children	71.80	60.45	77.00	80.40	73.50	71.10
CD45RO^+^CD27^−^CD8^+^	Mothers	36.20	38.25	51.20	34.10	39.70	37.00
	Children	28.20	39.55	23.00	19.60	26.50	28.90
CD45RO^+^CD57^+^CD8^+^	Mothers	50.40	42.20	50.60	35.00	50.50	41.70
	Children	43.25	60.20	45.80	35.70	44.50	48.30
CD45RO^+^CD57^−^CD8^+^	Mothers	49.60	56.00	49.40	65.00	49.50	57.30
	Children	56.75	39.80	54.20	64.30	55.50	51.70
CD45RO^+^CD57^+^CD27^+^CD8^+^	Mothers	8.21	4.40	4.09	2.32	7.11	4.18
	Children	3.34	2.65	3.13	3.87	3.21	2.96
CDRO^+^CD57^−^CD27^−^CD8^+^	Mothers	4.97	6.12	9.64	13.20	8.93	7.72
	Children	2.23	3.80	2.60	2.85	2.51	3.25

* Mothers developed Incident ≥ CIN+ during the follow-up. *p*-values = ^a^ 0.019, ^b^ 0.006, ^c^ 0.008, ^d^ 0.018, ^e^ 0.033.

**Table 5 viruses-14-02633-t005:** The proportion of CD4^+^ or CD8^+^ lymphocytes by their immunophenotypic subsets * given as mean (±SD) percentages among the mothers with persistent genital or oral HPV16 HPV-negative mothers. Only lymphocyte subsets with statistically significant differences are given, results are bolded.

	Genital HPV16 Infection		Oral HPV16 Infection		Combined HPV16 Infection	
	Persistent Infection (*n* = 10)	Always Negative (*n* = 20)	Persistent Infection (*n* = 7)	Always Negative (*n* = 5)	Persistent Infection (*n* = 17)	Always Negative (*n* = 25)
	**Mean (±SD)**		**Mean (±SD)**		**Mean (±SD)**	
Lymphocytes (CD3^+^)	69.03 (14.17)	56.76 (11.40)	48.17 (7.35)	60.26 (14.26)	59.91 (15.59)	57.46 (11.78)
**CD4^+^ cell population**						
HLADR^+^CD3^+^ **	5.67 (3.33)	5.17 (2.47)	**4.07 ^a^ (1.53)**	**6.60 ^a^ (2.15)**	4.97 (2.74)	5.46 (2.44)
**CD8^+^ cell population**						
CD45RO^+^CCR7^−^	**90.19 ^b^ (4.97)**	**80.56 ^b^ (13.40)**	88.47 (7.48)	88.12 (2.62)	**89.44 ^c^ (6.03)**	**82.07 ^c^ (12.36)**
CD38^+^HLADR^+^CD3 ***	3.92 (2.44)	5.40 (4.38)	**3.67 ^d^ (2.18)**	**7.70 ^d^ (3.61)**	3.81 (2.26)	5.86 (4.27)
CD45RA^+^CCR7^−^	92.96 (3.18)	90.21 (5.63)	95.37 (2.15)	94.14 (0.55)	**94.01 ^e^ (2.97)**	**91.00 ^e^ (5.26)**

*p*-values = ^a^ 0.038, ^b^ 0.048, ^c^ 0.033, ^d^ 0.036, ^e^ 0.044. * Percentages of CD4^+^ and CD8^+^ T-cell subpopulations. PBMCs were analyzed by flow cytometry by first gating on the total PBMCs and then on the CD3^+^ T cells. ** Markers expressed as percentages of total CD3 positive CD4 lymphocytes. *** Markers expressed as percentages of total CD3 positive CD8 lymphocytes.

**Table 6 viruses-14-02633-t006:** Proportion of CD4^+^ or CD8^+^ lymphocytes by their immunophenotypic subsets * given as mean (±SD) percentages among children whose mothers had persistent genital or oral HPV16 infection or were always HPV-negative. Only the lymphocyte subsets that had a statistically significant difference between the groups are given, results are bolded.

	Mother’s HPV16 Status					
	Genital HPV16 Infection		Oral HPV16 Infection		Combined HPV16 Infection	
	Persistent * Infection (*n* = 10)	Always HPV-Negative (*n* = 8)	Persistent * Infection (*n* = 7)	Always HPV-Negative (*n* = 3)	Persistent * Infection (*n* = 17)	Always HPV-Negative (*n* = 11)
	**Mean (±SD)**		**Mean (±SD)**		**Mean (±SD)**	
**Lymphocyte Subsets In Children**						
Lymphocytes (CD3^+^)	67.82 (13.99)	72.49 (13.67)	57.99 (12.77)	65.30 (10.67)	63.77 (14.01)	70.53(12.84)
**CD4^+^ cell population**						
CD38^+^HLADR^+^CD3^+^ **	4.86 (8.11)	2.65 (1.08)	**3.22 ^a^ (1.32)**	**1.24 ^a^ (0.36)**	4.18 (6.19)	2.26 (1.13)
**CD8^+^ cell population**						
HLADR^+^CD3^+^	9.45 (9.60)	7.99 (5.12)	**9.79 ^b^ (2.42)**	**4.17 ^b^ (1.05)**	9.59 (7.35)	6.95 (4.67)
CD38^+^HLADR^+^CD3^+^ ***	6.40 (9.03)	4.46 (2.76)	**6.44 ^c^ (1.62)**	**3.06 ^c^ (0.35)**	6.41 (6.85)	4.08 (2.41)
CD38^−^HLADR^+^CD3^+^CD4^+^	11.92 (25.21)	4.19 (3.00)	**4.25 ^d^ (1.49)**	**1.52 ^d^ (0.70)**	8.76 (19.33)	3.46 (2.82)
CD45RO^+^	40.81 (17.42)	38.59 (8.58)	**34.30 ^e^ (6.90)**	**23.40 ^e^ (2.82)**	38.13 (14.12)	34.45(10.17)

*p*-values: ^a^ 0.038, ^b^ 0.005, ^c^ 0.0080 ^d^ 0.018, ^e^ 0.033. * Percentages of CD4^+^ and CD8^+^ T-cell subpopulations. PBMCs were analyzed by flow cytometry by first gating on the total PBMCs and then on the CD3^+^ T cells. ** Markers expressed as percentages of total CD3 positive CD4 lymphocytes. *** Markers expressed as percentages of total CD3 positive CD8 lymphocytes.

## Data Availability

Data and materials of this study are available from the corresponding author upon request.

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
