# Peer review of "Peripheral Blood T-lymphocyte Phenotypes in Mother-Child Pairs Stratified by the Maternal HPV Status: Persistent HPV16 vs. HPV-Negative: A Case-Control Study"

_viruses, 2022, doi:10.3390/v14122633_

Round 1
Reviewer 1 Report
The findings appear to be novel and clinically relevant. Minor suggestions for improvement are as follows:
1. The number of subjects should be increased if possible to obtain more conclusive results.
2. Why the mean rank distribution of CD8+ memory cells was higher?
3. Why the median levels of both CD4+ and CD8+ cells was lower? The molecular mechanisms should also be evaluated in detail.
4. Why higher levels of activated CD4+ and CD8+ and circulating CD8+ memory cells were found among children?
5. Minor typographical errors were found throughout the manuscript and should be corrected.
6. The discussion section should be elaborated and improved.
Reviewer 2 Report
The article deals with the phenotypes of peripheral T cells in mothers who have persistent HPV16 infections and their children. The research topic is interesting; however, it contains statistical problems making the findings difficult to judge.
1. It seems that 138 t-tests have been performed in table 3. With a p<value set at 0.05, close to 7 significant differences could be found by chance in table 3. In table 3, 3 significances were found. This is also the case in table 4 where even more t-tests seem to have been performed. Correction for multiple testing is needed to be able to know if the findings are valid or not also considering the low number of patients being examined.
2. In methods the following is stated “Persistent “HPV16 infection (genital or oral) was defined by testing HPV16-positive at least in two (or more) consecutive follow-up visits”. The authors write in Discussion “Similarly, the temporal relationship between HPV acquisition and collection of the PBMCs is not known. Thus, the dynamics of adaptive immunity cannot be thoroughly investigated from these data.”
Were all women with persistent infections positive at baseline? If all included women with persistent infections were positive at baseline, it is difficult to relate the blood test to when the HPV infection was acquired, however, if an incident infection occurred later in follow-up, it is possible to relate to a time frame of acquisition of infection. This could be of interest to relate to the immune profile.
3. Are only persistent oral HPV infections in mothers reflected by a change in immune cells in their children? The authors conclude that persistent oral HPV16 infections in mothers is reflected by a down-regulated immune system in their children. However, standard deviations are much larger for immune cells taken in children of mothers with persistent cervical HPV16 infections than in persistent oral HPV16 infections. I would recommend that this is discussed in the manuscript. If the authors have reasons to believe that there is a difference in the immunological response in children depending on site of HPV infection, this should also be discussed in the manuscript.
4. What do we know of other concomitant infections in mothers? Could this be reflected in the immunological responses in mothers and their children?
Minor comments:
1. Line 189: Please change:” The same was true”.
Round 2
Reviewer 2 Report
The problem of reaching significant p-values by chance is valid regardless of if the table is divided into 3 factor variables (e.g., HPV status in different mucous membranes) and 25 T cell categories or 25 factor variables and 3 T cell categories. This is avoided by limiting the number of t-tests performed by limiting the number of hypotheses or compensating for multiple testing. To note, some of the significant findings are also in contrast with previous larger studies. In some of the studies referred to in the manuscript (Rodriguez et al, 2011) with similar calculations, the problem of multiple testing is also discussed. Therefore, statistical revision is needed. Another option-most likely not possible due to the unique cohort of patients- to validate significant findings in new cohorts.
